# Unifying host-associated diversification processes using butterfly–plant networks

Mariana P. Braga [1], Paulo R. Guimarães Jr[2], Christopher W. Wheat[1], Sören Nylin[1] & Niklas Janz[1]

Explaining the exceptional diversity of herbivorous insects is an old problem in evolutionary ecology. Here we focus on the two prominent hypothesised drivers of their diversification, radiations after major host switch or variability in host use due to continuous probing of new hosts. Unfortunately, current methods cannot distinguish between these hypotheses, causing controversy in the literature. Here we present an approach combining network and phylogenetic analyses, which directly quantifies support for these opposing hypotheses. After demonstrating that each hypothesis produces divergent network structures, we then investigate the contribution of each to diversification in two butterfly families: Pieridae and Nymphalidae. Overall, we find that variability in host use is essential for butterfly diversification, while radiations following colonisation of a new host are rare but can produce high diversity. Beyond providing an important reconciliation of alternative hypotheses for butterfly diversification, our approach has potential to test many other hypotheses in evolutionary biology.

[1] Department of Zoology, Stockholm University, Stockholm 10691, Sweden. [2] Departamento de Ecologia, Instituto de Biociências, Universidade de São Paulo, São Paulo, SP 05508-900, Brazil. Correspondence and requests for materials should be addressed to M.P.B. (email: mariana.braga@zoologi.su.se)

The diversification of herbivorous insects is one of the most successful animal radiations in the history of life[1], hence understanding its drivers is central to understanding a major mode of evolution. Ever since Ehrlich and Raven[2] argued for interactions between herbivorous insects and their host plants as being central to the diversification of both, and in the process formalising the concept of coevolution, evolutionary ecologists have searched for evidence of how such interactions could drive diversification[3,4]. Ehrlich and Raven assumed that a trait that allows an individual organism to explore a novel niche also promotes diversification, as the new niche would equate to a new adaptive zone, relatively free from competition. However, the mechanism connecting the increase in individual fitness to an increase in cladogenesis was not specified[5]. This gap in how micro- and macroevolution are connected has resulted in a range of proposed mechanisms linking insect–plant interactions to diversification[6–11]. Unfortunately, to date no clear consensus has emerged regarding the relative importance of these mechanisms. Here, we seek to advance this debate by reconciling the two most prominent and opposing explanations for the evolution of insect–plant interactions. We do so by proposing an approach to disentangle evolutionary hypotheses based on the patterns of interaction they are expected to produce.

The colonisation of a new host plant is often recognised as an opportunity for insect diversification. The various hypotheses of how and in which cases colonisation leads to diversification can be placed along two main axes: (i) the relative prevalence of complete host shifts vs. expansion of the number of hosts, which depends on variability in the insect host range, and (ii) the relative importance of key innovations vs. existing abilities (standing genetic variation and phenotypic plasticity) for colonisation of new hosts. Here we compare two alternative extremes among the above-mentioned explanations, herein referred to as the adaptive radiation scenario and the variability scenario. Each scenario aims to explain how changes in host use affect net diversification rates (without, however, teasing apart speciation, and extinction rates). The adaptive radiation scenario hypothesises that herbivorous insects quickly radiate into many species following a shift from an old to a novel plant taxon, by overcoming their host defences. As such, this is consistent with the idea of a key innovation by Ehrlich and Raven, though it does not require subsequent coevolution[7]. Rather, it is the complete change in host use, which increases the chances for ecological and geographic divergence, that are considered the main drivers of insect diversification[8]. In contrast, the variability scenario predicts that diversification is maximised in insect taxa with large variability in host use (aka the plasticity scenario[7,12]). Such variability results from the mixing and matching of hosts acquired by generalist ancestors and retained in the fundamental host repertoire (analogous to fundamental niche). Although most descendant species specialise on a subset of the ancestor's host repertoire, they retain the ability to use a wider range of potential hosts, including taxonomically distant plant taxa. The existence of such potential hosts—remnants of past host range expansions—makes host ranges unstable over evolutionary time, as insects can mix and match between hosts relatively easily. The resulting oscillations in host range increase the chance of population fragmentation and thereby speciation, via both adaptive and neutral processes[7].

Distinguishing between the radiation and variability scenarios is extremely challenging, as the complexity of host use makes it intractable for most phylogenetic reconstruction methods[3]. Most phylogenetic methods can reconstruct either the association between a given insect group and one host plant taxon at a time (and then combine the inferences from taxon-specific models; e.g.[13]), or the evolution of host range per se without specifying host taxa. Although there is an increased realisation that host

range is labile across time and space[13–16], its importance for diversification of herbivorous insects is still under debate[17–19]. Novel statistical approaches to study state-dependent diversification have been developed recently[20,21], but have so far produced divergent results and, consequently, different explanations for the effect of host range on diversification[10,11,17]. Part of this problem arises from the classification of host range, which is a complex trait, into two opposing states (specialist vs. generalist) or multiple states. Strictly speaking, host range is not an independently evolving trait, but rather an emergent property of the underlying dynamics of gaining and losing specific host plant taxa.

To investigate the role of hosts in diversification processes one would thus need to incorporate both the number of hosts used by each taxon (i.e., host range) and the identity of the host plants (i.e., host repertoire). A challenge to solve is how to circumvent computational limitations that constrain the application of such a method when modelling the evolution of host use. An alternative solution for this problem is to contrast the different patterns of interaction between insects and their host plants predicted by different diversification processes. Network analysis is a promising approach for this purpose[22], as it provides not only a visual representation of complex ecological systems, but also a formal way to quantify patterns of interaction in the studied system[23]. The mechanisms underlying these patterns can then be assessed using independent sources of information, such as phylogenetic relationships[24–27].

The butterfly families, Nymphalidae and Pieridae, were two of the examples of coevolution used by Ehrlich and Raven[2], and today are the primary examples of the variability and radiation scenarios, respectively. Nymphalidae comprises much of the diversity of butterflies and also shows dramatic variability in host use. The variability scenario was first proposed based on host use patterns in this family[28], but the diversification of at least one tribe, Satyrini, seems to be a radiation on a novel host clade[29,30]. Diversification of Pieridae, on the other hand, has been viewed by many as adhering to the adaptive radiation scenario[2], wherein radiation of the Pierinae followed the colonisation of the chemically well-defended Brassicales host plants. Later studies found support for such a butterfly–plant arms race[9,31].

Here, we estimate the relative importance of the radiation and the variability scenarios by translating their predictions into network properties (see Results) and investigating these processes in the butterfly families Nymphalidae and Pieridae. As the diversification of nymphalid and pierid butterflies are often seen as classic examples of the variability scenario vs. the adaptive radiation scenario, respectively, we expected to find contrasting patterns of interaction between these butterflies and their host plants. Instead, although network structure varies between the two groups, the patterns of interaction in both families have much in common, leading us to propose a unified explanation for the evolution of butterfly–plant interactions. The proposed approach thus appears to be a promising tool to assess whether the same dynamics apply to host–parasite systems in general, and to evaluate other hypotheses about evolutionary dynamics and diversification.

## Results

**Diversification scenarios and network structure**. Here we represent butterfly–plant interactions as a network, with each taxon (butterfly or plant) being a node and connections between nodes arising from their interaction (i.e., host–plant usage). In this network, butterflies using plants in the same family can then be clustered by this shared connection. Thus, if most of the diversity of butterflies was generated by adaptive radiations on

new host plants, the resulting network should be highly modular. Modularity emerges when a network contains recognisable sub-sets of taxa that interact more with each other than with other taxa in the network. Each module would then be composed of closely related plant taxa, which represent a distinct adaptive zone, and closely related butterflies, which descend from the ancestor that made the host shift. On the other hand, the varia-bility scenario would produce a nested butterfly–plant network. Nestedness emerges if (i) there is a specialist-generalist gradient in both trophic levels and (ii) the interacting assemblage of a taxon is a subset of the interacting assemblages of taxa with more interactions. In the variability scenario, temporal changes in host range produce a specialist-generalist gradient at any point in time, with specialised species utilising a subset of the host plants of their closely related generalists, which creates network nestedness.

To validate these predictions, we used a fixed tree and simulated butterfly diversification as taking place owing to either the radiation or the variability scenario, or various combinations

of the two (Fig. 1, see Methods and R code in the Supplementary Software for details). The tree was composed of 100 terminal taxa, separated into 10 clades grouped in pairs, with each pair having subclades with a low and high number of taxa ($n = 5$ and 15 taxa, respectively; Fig. 1a). This way, the difference in diversity between subclades in each pair could be generated by either one of the diversification scenarios. For comparison with a neutral scenario, we also simulated a network by randomly choosing 20% of the butterfly–plant interactions. For each simulation, we then analysed the resulting butterfly–plant network to see how well we could detect the relative contributions of the two scenarios that were simulated (Fig. 1b, Supplementary Table 1). We also recorded the number of hosts used by each butterfly taxon to compare with empirical networks (Supplementary Fig. 2).

According to our expectations, when diversification in all five clade pairs was generated by the radiation scenario (R5V0 in Fig. 1), the network was highly modular, with each module being composed of closely related butterflies and one plant taxon. As we

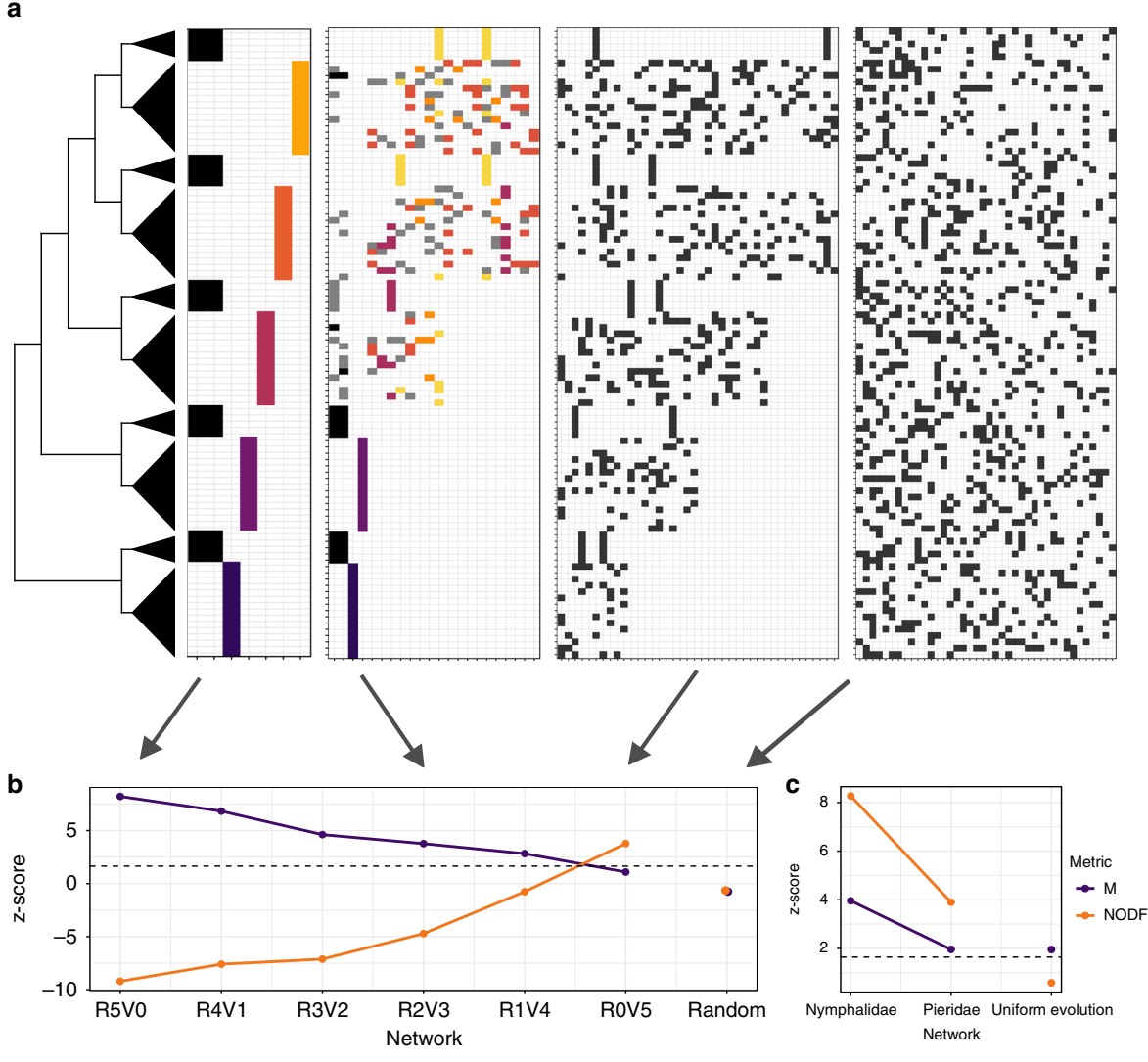

**Fig. 1** Structure of networks across simulated scenarios. **a** Phylogenetic tree of hypothetical butterfly group where triangle size is proportional to clade diversity. Four of the simulated networks, where each coloured cell represents an interaction between a butterfly taxon (rows—ordered to match the phylogeny) and a host taxon (columns). Interactions are coloured by module when network is significantly modular. **b** Z-score for nestedness (NODF) and modularity (M) for each simulated network, calculated by comparing each estimate to null model expectations (see Methods for more details). The horizontal dashed line indicates the significance threshold for values higher than expected under the null model. Simulation R5V0 represents the pure radiation scenario (i.e., five radiations) and R0V5 represents the pure variability scenario, with intermediate levels of both in between them. "Random" is the network where interactions were assigned randomly. **c** Z-score for empirical networks and simulated network produced by the uniform evolution scenario

decreased the number of diversification events by adaptive radiations, replacing them with diversification by variability in host use (R4V1–R1V4), network modularity decreased, but was still higher than expected by the theoretical benchmark provided by null model (see Methods). Even when diversification in only one of the five pairs followed the radiation scenario (R1V4) the network was still modular and not nested. This pattern shifted when diversification in the whole phylogeny was generated by the variability scenario (R0V5), which produced a nested and not modular network. We interpret these results as indicating that forming modules is much easier than creating nestedness, as the latter does not readily emerge from such simulations. These results suggest that with real data from much larger clades, detecting modules produced by the radiation scenario will be easier than detecting nestedness produced by the variability scenario. Finally, when interactions are randomly chosen, the levels of modularity and nestedness were not significant (Random in Fig. 1b and Supplementary Table 1) and the number of hosts used per butterfly followed a binomial distribution (Supplementary Fig. 2a).

**Butterfly–plant network structure.** To quantify the nestedness and modularity of butterfly–plant interactions, we constructed presence/absence matrices of interactions using existing literature (Methods). The Nymphalidae-plant network included 566 interactions between 295 Nymphalidae genera and 43 host–plant orders, and the Pieridae-plant network included 126 interactions between 67 Pieridae genera and 34 host–plant families. For consistency between butterfly families with respect to the classification level of plants, we also analysed a network between Nymphalidae genera and plant families. Nymphalidae network structure is very similar at both order and family level (see Supplementary Methods, Supplementary Figs. 3 and 4). For the other analyses we focused on the network at order level because that is the taxonomic level at which ancestral-state reconstructions of host use have been done for Nymphalidae.

For each network, we analysed nestedness[32] and modularity[33]. The Nymphalidae-plant network is both more nested (NODF = 13.09, permutation test, $p < 0.001$, z score = 8.27) and modular ($M = 0.58$, permutation test, $p < 0.01$, z score = 3.96) than networks generated by the null model (Fig. 1c). Butterflies and plants were grouped in 10 modules by an optimisation algorithm that maximises modularity (Figs. 2 and 3a). The smallest module, M7, has only four taxa (two butterfly genera and two plant orders) and is the only module that has no interactions with other modules of the network. The remaining nine modules are formed by at least 20 taxa, which are connected by one of the nine main host–plant orders (module and network hubs in Fig. 3c). In addition to nestedness at network level, within-module interactions are also significantly nested in two modules (M1: NODF = 49.89, permutation test, $p = 0.03$; M6: NODF = 60.21, permutation test, $p < 0.001$).

Contrary to our expectations, the Pieridae-plant network is also both significantly nested (NODF = 14.23, permutation test, $p < 0.001$, z score = 3.9) and modular ($M = 0.66$, permutation test, $p = 0.03$, z score = 1.96; Fig. 1c). This network is structured in 10 modules, three of them composed by only one butterfly–plant interaction (Figs. 3b and 4). Butterflies and plants in the four modules with more than 10 taxa are connected by the main plant family in the module, or module hub (Fig. 3d). Within-module interactions are also nested in two of these modules (M7: NODF = 63.13, permutation test, $p = 0.02$; M8: NODF = 72.14, $p < 0.001$). Although both networks have the same number of modules, the Pieridae-plant network has fewer interactions

between modules (16.6% of interactions) than the Nymphalidae-plant network (28.8% of interactions).

As the empirical networks show signs from both diversification scenarios, we simulated an additional scenario to test whether nestedness and modularity could have emerged simply from phylogenetic signal in the repertoire of hosts used by butterflies. In this scenario (herein referred to as uniform evolution, Supplementary Fig. 1b), the fundamental host repertoire evolved uniformly along all branches of the butterfly tree, resulting in fundamental host repertoires of the same size at the tips of the tree. Closely related clades shared more hosts, whereas basal clades had more unique hosts. Importantly, low and high-diversity subclades within each of the five pairs of clades had the same fundamental host repertoire. Then, realised repertoires were randomly sampled from the fundamental host repertoire. The resulting network was not significantly nested (NODF = 10.55, permutation test, $p = 0.26$, z score = 0.59), but modularity was slightly higher than expected by the null model ($M = 0.54$, permutation test, $p = 0.02$, z score = 1.96; Fig. 1c; Supplementary Table 1). Thus, phylogenetic conservatism in host repertoire alone can create low levels of modularity, but for nestedness to emerge, phylogenetic conservatism has to be coupled with host range expansion events (as in the variability scenario).

Comparing the simulated and empirical networks, the high levels of nestedness in the empirical networks suggest that the variability scenario played an important role on the diversification of both butterfly families. The modularity levels, however, could have emerged simply from phylogenetic conservatism in host repertoire, especially in Pieridae, where modularity is low.

**Structural roles of ancestral and recent hosts.** According to the variability scenario[7], the pool of host plants used by a clade derives mainly from previous events of polyphagy. Recent reconstructions of past host use for nymphalid butterflies suggest Rosales as the most likely ancestral host order followed by Malpighiales, which suggests that both orders were used by a generalist ancestor early in the evolution of the family[13]. These plant orders are probably the ones with the longest evolutionary association with nymphalid butterflies, and therefore may have an important role in shaping structural patterns of the studied network. Besides the ancestral hosts, two recent host orders—Poales and Solanales—support species-rich butterfly taxa, which are likely the result of radiation events[29,34]. Hence, we expect these hosts to also have an important effect on network structure.

In support of the proposed link between diversification scenarios and network structure, we found that ancestral hosts produce nestedness and recent hosts produce modularity in the Nymphalidae-plant network. Nestedness is significantly lower when Rosales and Malpighiales are removed from the network, as compared with the effect of all other host plants (NODF = 11.39, permutation test, $p = 0.002$, z score = $-1.67$), whereas modularity decreases most when Poales and Solanales are removed ($M = 0.52$, permutation test, $p = 0.001$, z score = $-5.14$).

Host use evolution in pierid butterflies is marked by a shift in host preference from Fabales (the probable ancestral host for all butterflies) to Brassicales. Because this shift was followed by increases in diversification rate[9], Brassicales plants are the most common hosts for pierids, especially from Capparaceae and Brassicaceae families. In the Pieridae-plant network Capparaceae and Brassicaceae have the strongest effect on network structure. Removal of these hosts significantly decreases nestedness (NODF = 11.38, permutation test, $p = 0.0018$, z score = $-6.25$) and increases modularity ($M = 0.72$, permutation test, $p = 0.0018$, z score = 5.31). Therefore, these hosts act as ancestral hosts that

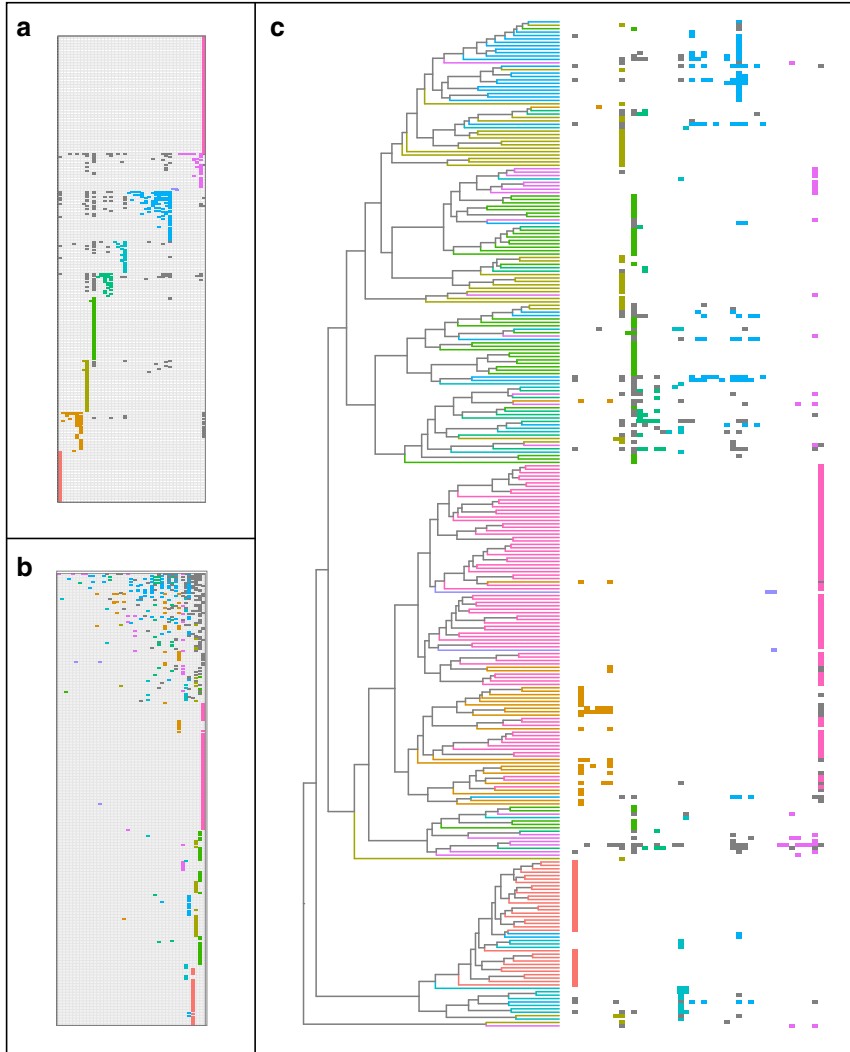

**Fig. 2** Three different ways of viewing the Nymphalidae-plant network. Butterfly genera are in rows and host plant orders are in columns. Each filled cell indicates a butterfly–plant interaction and each colour shows interactions within a module (grey cells are interactions between modules). **a** Rows and columns sorted to emphasise modular affinity (order of modules is arbitrary). **b** Rows and columns sorted to emphasise nestedness, ordered from the upper right corner according to descending number of interactions. **c** Resorting of the rows of the matrix (right) to match the Nymphalidae phylogeny (left) and highlight the phylogenetic diversity in each module. Branches of the phylogeny are coloured by module affiliation of the terminal taxa. Note that module colours are consistent with Fig. 3. For taxon names see Supplementary Figs. 5 and 6

promote variability in host use, despite having supported butterfly radiations in the beginning of this ecological association.

**Phylogenetic composition of modules**. Most modules with more than five butterfly taxa in both networks are composed of phylogenetically closely related butterflies (Tables 1 and 2, Figs. 2 and 4). The two exceptions are modules with butterflies specialised on ancestral hosts: M2 of the nymphalid network, which includes Rosales, and M3 of the pierid network, which includes Capparaceae. These exceptions support the expectation from the variability scenario that butterflies retain the ability to use ancestral hosts. As for the phylogenetic diversity of plant orders, with the exception of the two modules that only have one host plant, all modules are composed of a phylogenetically widespread combination of host plants (Tables 1 and 2). These results indicate that host use is phylogenetically conserved (related butterflies use the same repertoire of plants), but this repertoire usually includes unrelated plant clades.

Combining our results, it is clear that the modular structure is formed by grouping closely related butterflies that use a main host taxon (module hub). But several modules also include a number of other distantly related hosts that are used by a subset of the butterflies in the module, producing nestedness within modules. Hosts with a long evolutionary history of association with the butterflies tie the various modules together, resulting in overall network nestedness.

## Discussion

Here, we describe and implement an approach to show that different host-associated diversification dynamics produce distinct butterfly–plant network structures. We then use this approach upon two of the original exemplar butterfly families that Ehrlich and Raven used to introduce coevolution[2], and despite the general acceptance that Nymphalidae and Pieridae underwent different diversification and host use processes during their evolution[9,28], we show that the network structures of the two families are very similar. We suggest that the evolution of butterfly–plant networks is mainly driven by the formation of new ecological interactions (initially at the population level, but carried over to the species level after speciation events), combined

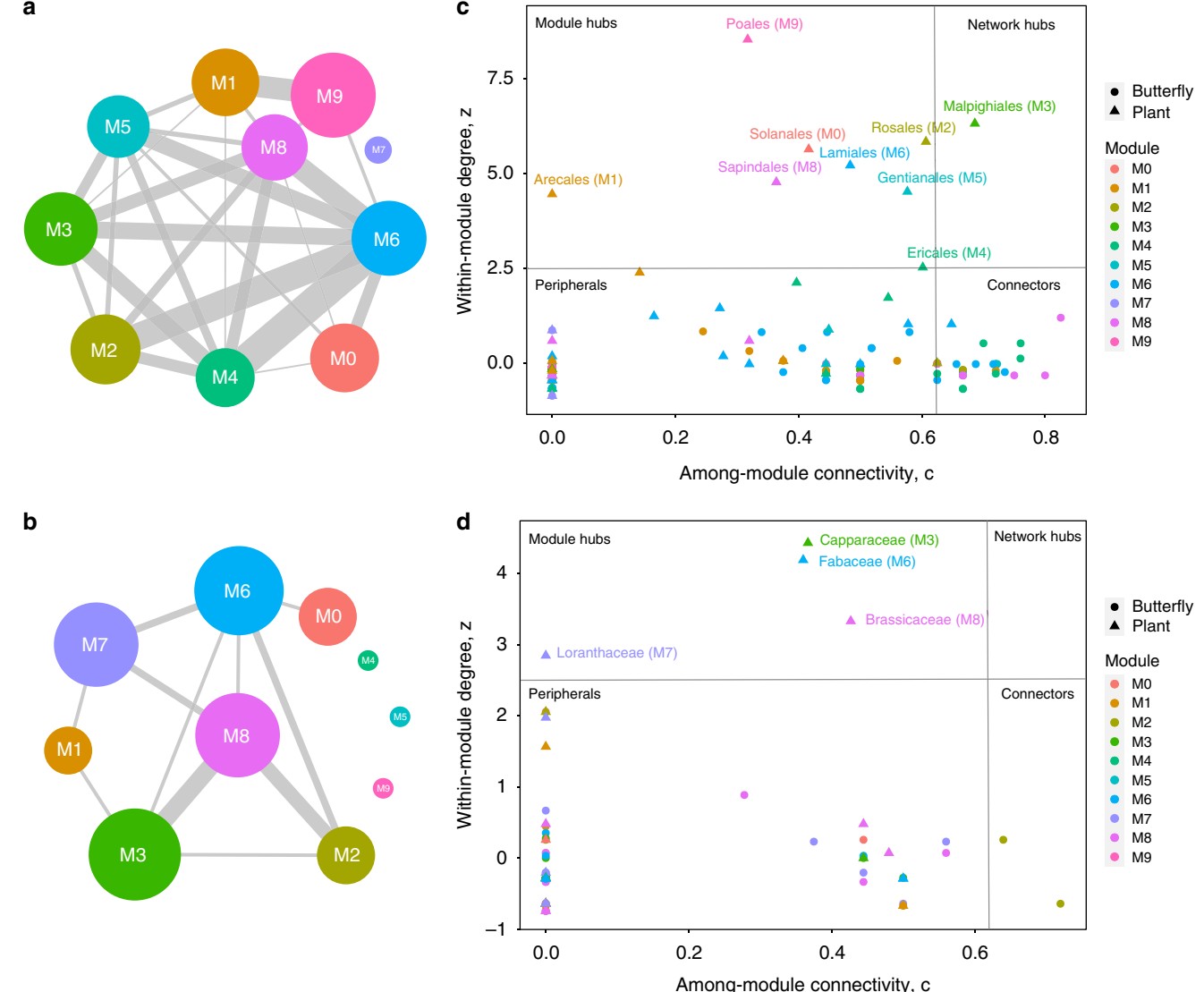

**Fig. 3** Overview of the modular structure and detailed contributions of taxa in the networks. **a, b** Graph of module interactions for the nymphalid and pierid networks, respectively. Circle size reflects the number of butterflies and plants within each module, and the width of the links (grey lines) between them reflects the number of interactions between modules. **c, d** Relative contribution of each taxon to the modular structure, defined by measures of connectivity with taxa within the same module (y axis) and with taxa assigned to other modules (x-axis) of the nymphalid and pierid networks. Names of modules and network hubs are shown. Colours of modules are consistent with Figs. 2 and 4

with phylogenetic conservatism. Conservatism in host use is one of the most prevalent characteristics of herbivorous insects[13,35], yet it does not prevent colonisation of new hosts when opportunity arises[36,37]. Instead, even highly specialised insects are expected to have a wider fundamental than realised host repertoire—analogous to fundamental and realised niche. Phenotypic plasticity, resource tracking, and recurrence homoplasy allow insects to continuously explore their fundamental host repertoire by probing new hosts[38]. Under some circumstances, this exploration produces patterns that can be detected in the network structure.

We suggest that the ubiquitous properties of network structure identified in this study reflect three phases in the evolution of butterfly–plant interactions. First, one of the host colonisations may lead to a complete shift in host use, especially if old and novel hosts are significantly different, as in the case of the shift from rosids to Poales (grasses) by Satyrinae, the largest Nymphalidae subfamily. The colonisation of Poales happened ~ 60 Mya by the common ancestor of Satyrinae + Morphini +

Brassolini[29], and the spread and diversification of Satyrini throughout the world happened ~ 40 Mya[30]. This diversification on grasses produced a clear butterfly–plant module with little plant phylogenetic diversity (M9 of nymphalid network). In general, we expect such events to be rare because colonisation of host groups that were not used before should be difficult and may not have much success in terms of diversification until specific traits (such as detoxification genes) evolve. And even in the case of a successful colonisation, not all novel plant groups would provide enough opportunities for diversification.

Second, herbivores continue to explore their fundamental host repertoire even after host shifts. For example, a complete shift in preference happened following the colonisation of Brassicales plants by Pieridae butterflies ~ 70 Mya[9] (which would represent the first phase). But, with time, pierids also colonised other host plants, breaking up the strict modular structure and creating nestedness within the module. In fact, it seems that the colonisation of the order Brassicales (first Capparaceae and subsequently, Brassicaceae) facilitated the colonisation of other plant

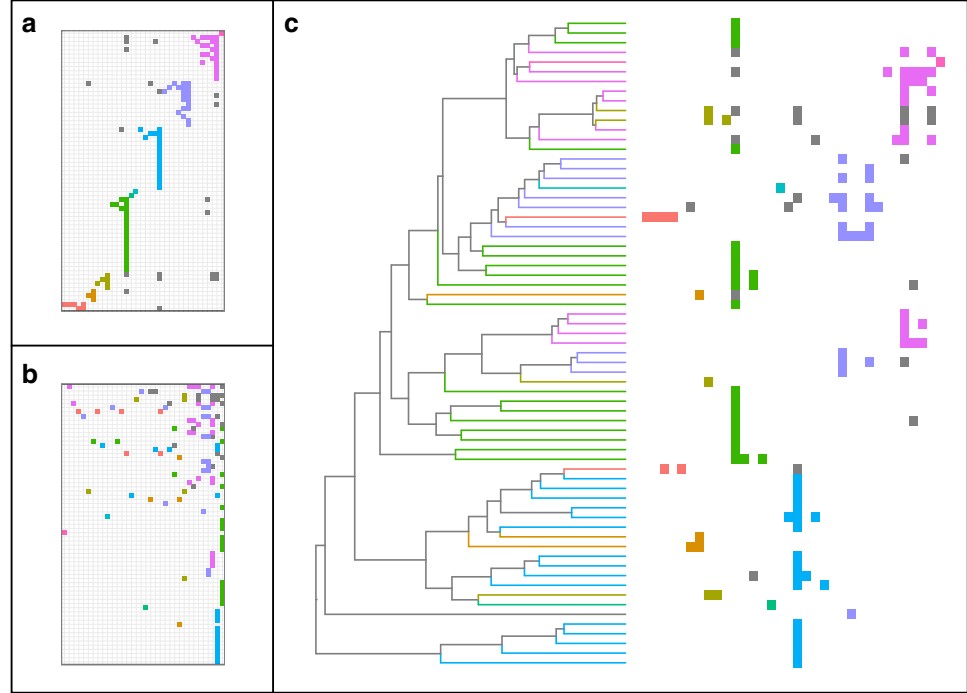

**Fig. 4** Three different ways of viewing the Pieridae-plant network. Butterfly genera are in rows and host plant families are in columns. Each filled cell indicates a butterfly–plant interaction and each colour shows interactions within a module (grey cells are interactions between modules). **a** Rows and columns sorted to emphasise modular affinity (order of modules is arbitrary). **b** Rows and columns sorted to emphasise nestedness, ordered from the upper right corner according to descending number of interactions. **c** Resorting of the rows of the matrix (right) to match the Pieridae phylogeny (left) and highlight the phylogenetic diversity in each module. Branches of the phylogeny are coloured by module affiliation of the terminal taxa. Note that module colours are consistent with Fig. 3. For taxon names see Supplementary Figs. 7 and 8

**Table 1 Phylogenetic diversity (PD) of butterflies and plants for each module on the Nymphalidae-plant network**

| Module | Butterflies | | | Plants | | |
|---|---|---|---|---|---|---|
| | #taxa | PD(%) | p | #taxa | PD(%) | p |
| M0 | 32 | **7.49** | **0.001** | 1 | 7.44 | – |
| M1 | 25 | **11.8** | **0.001** | 6 | 21.07 | 0.274 |
| M2 | 33 | 17.66 | 0.223 | 2 | 9.5 | 0.465 |
| M3 | 40 | **16.47** | **0.001** | 2 | 11.57 | 0.796 |
| M4 | 15 | **8.03** | **0.003** | 5 | 17.36 | 0.172 |
| M5 | 20 | **10.47** | **0.002** | 4 | 19.01 | 0.685 |
| M6 | 32 | **14.38** | **0.001** | 13 | 38.84 | 0.261 |
| M7 | 2 | 1.63 | 0.29 | 2 | 14.88 | 0.973 |
| M8 | 22 | **11.52** | **0.004** | 7 | 26.45 | 0.454 |
| M9 | 74 | **25.54** | **0.001** | 1 | 7.44 | – |

Assemblages with lower phyloglgenetic diversity than expected by chance are in bold

families. These include distantly related Brassicales families such as Tropaeolaceae, but also plants from entirely unrelated orders, such as the family Loranthaceae (showy mistletoes) from the order Santalales. In other words, colonisation of Brassicales led to an increase in variability in host use in pierid butterflies, producing within-module nestedness, and characterising a second phase of host use evolution.

Third, the recurrent addition of new host plants increases among-module interactions. The pierid network is surprisingly similar to the nymphalid network. The main difference is that Nymphalidae is a larger clade with more variability—but also overlap—in host use, which is reflected in network size and inter-module interactions. These differences might be partially explained by the time of association between the butterflies and the main ancestral host group. The association between

nymphalid butterflies and their ancestral host, Rosales, is ~ 100 million years old[13,39], the oldest one in this study, whereas the pierid association with Brassicales is ~ 70 million years old[9]. Although the ability to use Rosales seems to be retained in most clades of Nymphalidae (high phylogenetic diversity in module M2), various clades use other host plants more often. These are not complete shifts in host plant, but changes in host use frequencies, and can be seen in the modular structure of the network, with highly connected modules (Fig. 3a). As a consequence, the third phase is characterised by both nestedness and modularity. Network nestedness (and therefore, coherence) is maintained by ancestral hosts, whereasmodularity increases with specialisation to new hosts (module hubs; Fig. 3c).

Here, our goal was to compare the main alternative hypotheses for host-associated diversification based on network properties

| Table 2 Phylogenetic diversity (PD) of butterflies and plants for each module on the Pieridae-plant network | | | | | | |
|---|---|---|---|---|---|---|
| | **Butterflies** | | | **Plants** | | |
| **Module** | **#taxa** | **PD(%)** | ***p*** | **#taxa** | **PD(%)** | ***p*** |
| M0 | 2 | 161.57 | 0.848 | 5 | 795.28 | 0.875 |
| M1 | 3 | 205.25 | 0.521 | 2 | 461 | 0.947 |
| M2 | 4 | 235.89 | 0.247 | 3 | 551.7 | 0.913 |
| M3 | 18 | 866.64 | 0.472 | 4 | 651.62 | 0.883 |
| M4 | 1 | 86.18 | – | 1 | 325.1 | – |
| M5 | 1 | 86.18 | – | 1 | 325.1 | – |
| M6 | 15 | **577.91** | **0.003** | 5 | 762.3 | 0.841 |
| M7 | 11 | **434.99** | **0.001** | 6 | 832.68 | 0.817 |
| M8 | 11 | **340.58** | **0.001** | 6 | 711.97 | 0.619 |
| M9 | 1 | 86.18 | – | 1 | 325.1 | – |

Assemblages with lower phylogenetic diversity than expected by chance are in bold

that emerge from the evolutionary dynamics. We focused on the colonisation process (or creation of new interaction in the network), which is the necessary first step for network assembly. Teasing apart the effects of speciation and extinction is a difficult task that requires being the focus of a future study. Although the two alternative extremes (radiation and variability scenarios) are indeed associated with opposite network properties (modularity and nestedness), the unification of these complementary parts results in a better description of host use evolution and diversification than each part separately. For instance, lack of variability in host use can be the reason why some colonisation events are not followed by rapid diversification (e.g., M7 of nymphalid network), and key innovations allowing colonisation of novel host taxa providing new niches can be thought of as evolutionary novelties that suddenly increase realised and fundamental host repertoires, and therefore the potential for variability in host use (e.g., M8 of pierid network).

In conclusion, we argue that the variability and radiation scenarios can be reconciled into a unified view of butterfly–plant evolution in which the continuous probing of new hosts allows both ongoing diversification through variability in host use and episodic radiations on new hosts. Somewhat ironically, this was foreshadowed already by Ehrlich and Raven in their seminal paper. In one passage that has been given much less attention than their arms-race coevolution ideas, they noted that "the degree of plasticity of chemoreceptive response and the potential for physiological adjustment to various plant secondary substances in butterfly populations must in large measure determine their potential for evolutionary radiation". With the recent recognition that host–parasite systems have much in common with herbivorous insect–plant systems[38], our approach could be applied to other host–parasite systems to test the generality of our conclusions. Moreover, we believe this study demonstrates the potential of using network analysis in a phylogenetic context to investigate hypotheses about macroevolutionary dynamics.

## Methods

**Diversification scenarios and network structure.** We simulated networks resulting from the evolution of host repertoire in a phylogenetic tree with five pairs of sister clades (Fig. 1a), where one clade contains five terminal taxa (low diversity clade) and the other contains 15 (high diversity clade). The difference in diversity in each pair was then associated with either the adaptive radiation or the variability scenario. Simulation of the radiation scenario followed three rules: (1) the ancestor of all clades used two hosts, (2) low diversity clades use the same two ancestral hosts, and (3) high diversity clades use a unique new host (which allowed the radiation). For the variability scenario the rules were: (1) the fundamental host repertoire of the ancestor includes 10 hosts, (2) 10 more hosts are added to the fundamental host repertoire of the more diverse clade after branching, (3) low diversity clades use two hosts randomly chosen from the fundamental host repertoire, and (4) in high diversity clades, 80% of all possible interactions between

terminal taxa and hosts in the fundamental host repertoire are removed with equal probability, which produces variation in specialisation.

Then six combinations were simulated (Supplementary Fig. 1), spanning from diversity in all five pairs explained by the radiation scenario (R5V0) to diversity in all pairs explained by the variability scenario (R0V5). For the intermediate networks, we started simulating the radiation scenario and shifted to the variability scenario at different points of the phylogeny (Supplementary Fig. 1d–g).

For comparison, we also simulated two networks where the evolution of host repertoire does not affect diversification. In the random network (Supplementary Fig. 1a), the fundamental host repertoire of all terminal taxa contained 40 hosts (same as in the variability scenario) and 20% of possible interactions (all combinations of butterflies and plants) were randomly chosen. Thus, realised host repertoires were randomly sampled from a fixed fundamental host repertoire. In the uniform evolution scenario, hosts were added to the fundamental host repertoire uniformly through time (Supplementary Fig. 1b), so that more closely related clades shared more hosts, whereas basal clades had more unique hosts. As in the random and the variability scenarios, 20% of possible interactions were randomly chosen. We then measured nestedness and modularity of each simulated network as described below.

**Butterfly–plant network structure.** To build the Nymphalidae-plant network we used the host use data set sampled by ref. [13], which is based on records of host plant orders for butterfly genera reported in the literature and on ref. [40]. We followed ref. [39] for the phylogenetic relationships between Nymphalidae genera, and for phylogenetic relationships between plant orders we followed refs. [41,42]. The interactions between Pieridae genera and plant families were also gathered from the literature[43–49]. We followed ref. [9] for the phylogenetic relationships between Pieridae genera, and refs [9,42] for phylogenetic relationships between plant families.

We used the program ANINHADO[50] to compute the NODF index, a nestedness metric based on overlap and decreasing fill[51]. To detect modularity, we used Newman and Girvan's metric[52] modified for bipartite networks[53] as implemented in the software MODULAR[54]. We used a simulated annealing algorithm to maximise the index of modularity ($M$) and identify the modules. As the algorithm is based on an optimisation process, the outcome of different runs may vary. That is particularly important for networks with many interactions between modules. Therefore, we ran the analysis 10 times and compared the resulting modules and index of modularity. As network configuration did not vary significantly across runs, we simply chose the one with highest modularity, $M$.

In order to produce null distributions of NODF and $M$-values, we computed these indices for 1000 matrices generated by a null model in which the probability of each interaction is proportional to the number of interactions of the insect and the plant, therefore taking into account heterogeneity in host range and in butterfly richness per host taxon (null model 2 of ref. [32]). Thus, if the observed patterns are significantly different from what is generated by the null model, such patterns do not emerge simply from a specialisation gradient but from another underlying process. Based on the null expectation, we then standardised NODF and $M$ values using Z-score $= \frac{X_{obs} - X_{exp}}{StDev_{exp}}$; where $X_{obs}$ is the metric of interest, $X_{exp}$ is the mean value and $StDev_{exp}$ is the standard deviation from the null distribution. This is a standardisation that quantifies the position of the observed metric within the null distribution in terms of units of standard deviation[55].

Based on topological properties, each taxon was assigned a role in the network following ref. [33]. The role of a node is defined by how it interacts within its own module (standardised within-module degree) and with nodes in other modules (among-module connectivity).

**Structural roles of ancestral and recent hosts.** We assessed the role of host plants by recalculating nestedness and modularity after removing one of all possible combinations of two plant taxa from the network. This resulted in 904 combinations for the nymphalid network and 561 combinations for the pierid

network. The importance of any given combination of hosts was assessed by calculating the Z-score for NODF and $M$-values of the network without the given combination of hosts in relation to all other networks.

**Phylogenetic composition of modules.** We calculated Faith's phylogenetic diversity[56] of butterflies and plants in each module and contrasted that to a null distribution, using the package picante version 1.6–2[57] of R[58]. Faith's phylogenetic diversity is the sum of the total phylogenetic branch lengths leading to the terminal taxa in the sample. We used a null model that shuffles taxon labels across tips of the phylogeny to generate expected values of phylogenetic diversity for each module, maintaining the number of butterflies and plants on each module.

**Code availability.** Custom code used to simulate theoretical diversification models is available as Supplementary Software.

## Data availability

The authors declare that the data supporting the findings of this study are available within the paper (and its Supplementary Information). A reporting summary for this Article is available as a Supplementary Information file.

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

## Acknowledgements

SN was supported by the Swedish Research Council (2015-04218) and PRG was supported by FAPESP (2016/20739-9) and CNPq. Discussions at the symposium "Changing species associations in a changing world: a Marcus Wallenberg symposium" (MWS 2015.0009 to SN) improved the manuscript.

## Author contributions

M.P.B, N.J. and S.N. conceived the study; S.N. and C.W.W provided the data; M.P.B and P.R.G designed the analyses; M.P.B analysed the data; M.P.B wrote the paper with input from all the other authors.

## Additional information

**Competing interests:** The authors declare no competing interests.

