## [Peer Review File · Nature Communications]

Reviewers' Comments:

Reviewer #1:

Remarks to the Author:

The revision is successful and I have no hesitation in seeing the paper move forward to publication. This one of the more interesting and challenging papers in the field of plant-insect macroevolution that I have seen in some years. I caught just a few typos and word issues, listed below.

Line 77. Maybe you could spell out what is meant by "combine them." Do you mean "...and then combine the inferences from taxon-specific models"?

117. Change to "apply".

135. Change to "produce".

273-274. The sentence "We have, therefore, performed..." seems more like something in a response to reviewers letter, I think it could be removed here.

285. Change "from" to "of".

355. Do you want "host preference" here or "host use", since the patterns you're looking at result from preference and ability to consume, etc.

Reviewer #2:

Remarks to the Author:

As before, I find the paper novel, interesting and worthwhile. I was also positively surprised by the extent to which all reviewers seemed to see eye to eye on the salient points of criticism – which the authors have now acted upon in a most commendable way. Excessive claims of generality have been adequately toned down and simulations further developed in way that adds even further to the value of this innovative paper.

One aspect that I only came to think of after submitting my previous review, and which I have been wondering about is the following: How do missing or lost taxa affect the inference? What I refer to is this: If you have different extinction rates in different lineages, will that not lead to different patterns among the remaining species? (For the sake of illustration, imagine the simplest of cases: you start from a well-filled interaction matrix and then start losing particular types of interactions (between given herbivore clades and given plant clades), then what will the "surviving" interaction matrix look like, and how informative will it be regarding previous dynamics along the route to the current matrix [I can imagine that it will be clumped, full of holes and potentially nested, no matter how host plants were originally colonized]? Somehow it would seem to me that the authors either assume that all species arising during evolutionary history are still here to be observed, or that extinction rates are constant and randomly distributed among taxa, and that extinction rates are independent of host use patterns. Yet we know (or I think that we know) that all three assumptions are false. And the simulation models seem to include no model of extinction, and not to explore this part of the equation in any way. Unfortunately, this aspect is way outside of my own expertise, but I would ask the authors to please put me straight by simply rebutting my concern with a simple explanation (and/or simulation)?

One added thing, which is mainly a matter of presentation and structure: the discussion includes some

in-depth discussion of the association between this and that taxon and this and that plant, to an extent that I find vastly exceeding what is actually needed to support the more general points addressed. I understand that a butterfly (or plant) enthusiast may find this truly cool, but given my own Spartan taste, I could very well do with less. Maybe consider removing some of the most system-specific inference to the appendix, where the true aficionados may still find it?

Reviewer #3:

Remarks to the Author:

This revised version is a nice piece of work. The authors have intergrated all my (and other reviewers') comments. I am particulartly happy with implementation of Figure 1.

We thank the editor and the reviewers for the efficient review process. As in the first round, all comments were relevant and contributed to the quality of the manuscript. All the issues raised by the reviewers are addressed below and the requested changes are highlighted in the main text. We hope you find the new version of the manuscript suitable for publication. Please let us know if you think we need to handle/clarify any additional point that might not be at the standards of Nature Communications yet.

Reviewer #1 (Remarks to the Author):

The revision is successful and I have no hesitation in seeing the paper move forward to publication. This one of the more interesting and challenging papers in the field of plant-insect macroevolution that I have seen in some years. I caught just a few typos and word issues, listed below.

Line 77. Maybe you could spell out what is meant by “combine them.” Do you mean “...and then combine the inferences from taxon-specific models”?

117. Change to “apply”.

135. Change to “produce”.

273-274. The sentence “We have, therefore, performed...” seems more like something in a response to reviewers letter, I think it could be removed here.

285. Change “from” to “of”.

355. Do you want “host preference” here or “host use”, since the patterns you’re looking at result from preference and ability to consume, etc.

RESPONSE: We truly appreciate the reviewer’s comments. We accepted all the suggestions and incorporated them to the main text.

Reviewer #2 (Remarks to the Author):

As before, I find the paper novel, interesting and worthwhile. I was also positively surprised by the extent to which all reviewers seemed to see eye to eye on the salient points of criticism – which the authors have now acted upon in a most commendable way. Excessive claims of generality have been adequately toned down and simulations further developed in way that adds even further to the value of this innovative paper.

One aspect that I only came to think of after submitting my previous review, and which I have been wondering about is the following: How do missing or lost taxa affect the inference? What I refer to is this: If you have different extinction rates in different lineages, will that not lead to different patterns among the remaining species? (For the sake of illustration, imagine the simplest of cases: you start from a well-filled interaction matrix and then start losing particular types of interactions (between given herbivore clades and given plant clades), then what will the “surviving” interaction matrix look like, and how informative will it be regarding previous dynamics along the route to the current matrix [I can imagine that it will be clumped, full of holes and potentially nested, no matter how host plants were originally colonized]? Somehow it would seem

to me that the authors either assume that all species arising during evolutionary history are still here to be observed, or that extinction rates are constant and randomly distributed among taxa, and that extinction rates are independent of host use patterns. Yet we know (or I think that we know) that all three assumptions are false. And the simulation models seem to include no model of extinction, and not to explore this part of the equation in any way. Unfortunately, this aspect is way outside of my own expertise, but I would ask the authors to please put me straight by simply rebutting my concern with a simple explanation (and/or simulation)?

RESPONSE: We thank the reviewer for appreciating our effort to revise the manuscript and we agree that it improved considerably the quality of the paper.

The problem of lost taxa or incomplete taxa sampling is for sure an important and complicated issue in macroevolutionary studies. We think, however, that this aspect is beyond the scope of this paper and do not imperil our findings because: (1) The main point of the study is to show how network analysis can be used to tease apart process-based hypotheses with different predictions in terms of network structure. However, the two hypotheses investigated in this paper do not have clear predictions about extinction rates. These hypotheses focus on net diversification rates without teasing apart the role of speciation and extinction (we now acknowledge that in the manuscript - line 56). That being said, even if we added heterogeneous extinction rates, (2) extinction by itself would not be enough to create the observed patterns. In the case of radiations, extinction (or loss of interactions) would probably reduce the connectance within modules (and therefore decrease modularity), but they would still be modules, since extinction can not create new interactions. On the other hand, in the variability scenario the butterflies have more plants to choose from, so there is a larger potential for extinction to create interaction patterns. In this case, the most commonly accepted idea is that specialist-specialist interactions are the most vulnerable, and that would increase nestedness, which would just reinforce the link we propose. To summarize, we focused on the colonization process (or creation of new interaction in the network), which is the necessary first step for network assembly. And even though we touch upon interaction loss (by sampling only 20% of the possible interactions in some of the modelled scenarios), teasing apart the effects of speciation and extinction is a difficult task that requires being the focus of a future study (or studies).

One added thing, which is mainly a matter of presentation and structure: the discussion includes some in-depth discussion of the association between this and that taxon and this and that plant, to an extent that I find vastly exceeding what is actually needed to support the more general points addressed. I understand that a butterfly (or plant) enthusiast may find this truly cool, but given my own Spartan taste, I could very well do with less. Maybe consider removing some of the most system-specific inference to the appendix, where the true aficionados may still find it?

RESPONSE: While we can relate to the reviewer's perspective, we would argue that the main reason for the examples given in the discussion is not to highlight how cool our butterflies are (though they are), but to back up our conclusions. With the examples explicitly stated, any reader is able to go back to the figures and match the names with the processes and patterns we are discussing. Also, it facilitates the connection between our system and whatever system the reader is interested in.

Reviewers' Comments:

Reviewer #2:

Remarks to the Author:

The authors have evaluated the last few suggestions and responded in a sensible and thoughtful way. I have no hesitation in recommending that the paper moves forward to publication. This study will make a great contribution to the literature.

Reviewer #3:

Remarks to the Author:

As I wrote earlier, I was really happy with the revision. On the specific point of the impact of extinction rates, I do agree with the authors that even by adding heterogeneous extinction rates, results would be very similar given that extinctions do not create interactions. That said if the editor feels it is important, the authors might be willing to add 1-2 sentences in the discussion with regard to this point, especially if the authors already have tried to include heterogeneous rates in their analysis. But once more, this should not impede publication of this nice piece of work.

REVIEWERS' COMMENTS:

Reviewer #2 (Remarks to the Author):

The authors have evaluated the last few suggestions and responded in a sensible and thoughtful way. I have no hesitation in recommending that the paper moves forward to publication. This study will make a great contribution to the literature.

Reviewer #3 (Remarks to the Author):

As I wrote earlier, I was really happy with the revision. On the specific point of the impact of extinction rates, I do agree with the authors that even by adding heterogeneous extinction rates, results would be very similar given that extinctions do not create interactions. That said if the editor feels it is important, the authors might be willing to add 1-2 sentences in the discussion with regard to this point, especially if the authors already have tried to include heterogeneous rates in their analysis. But once more, this should not impede publication of this nice piece of work.

Following the advice of reviewer 3 and the editor, we now mention the issue raised about extinction and interaction loss in the discussion.